# Trialling the efficacy of a technological visuo-cognitive training program as a compensatory tool for visual rehabilitation after stroke: A pilot study

Lewis Jefferson[1], Abbey Fletcher[2], Beckie Morris[1], Julia Das[3], Rosie Morris[3], Samuel Stuart[3], Stephen Dunne[1]*

1 Department of Psychology, Faculty of Health and Wellbeing, Northumbria University, Newcastle upon Tyne, United Kingdom, 2 School of Psychology, University of Sunderland, Sunderland, United Kingdom, 3 Department of Sport, Exercise and Rehabilitation, Northumbria University, Newcastle upon Tyne, United Kingdom

* stephen.dunne@northumbria.ac.uk

## Abstract

Visual impairments are common post-stroke and can lead to diminished functioning and difficulty accomplishing everyday tasks, such as reading and navigating unfamiliar environments independently. This pilot study investigates the usability, acceptability and preliminary efficacy of technological visuo-cognitive training (TVT) using the Senaptec Sensory Station for stroke survivors with visual field loss. Ten stroke survivors (8 males, 2 females; 43–79 years old; $M_{age} = 65$, $SD_{age} = 11.03$) with a non-progressive visual field defect underwent TVT comprising baseline assessment, five 30-minute training sessions over 2–3 weeks, and post-intervention assessment. Measures of visual cognition, patient-reported outcomes, usability, and acceptability were assessed pre- and post-intervention, supplemented by qualitative interviews. Participants demonstrated meaningful gains in several aspects of visual search and functional vision. Reaction times on target capture tasks improved significantly, mirrored by more efficient performance on the Bell's Test. These behavioural changes aligned with reductions in reported visual difficulties and fatigue, both showing large effect sizes. Across sessions, participants also showed improvement in hand–eye coordination and visuomotor integration. Engagement with the system was high: perceived competence increased and usability ratings were excellent. Qualitative accounts contextualised these findings, describing enjoyment of the technology, occasional challenges related to adaptive difficulty or physical limitations, and perceived benefits such as greater awareness of visual scanning strategies in daily life. Notably, several sensory measures (e.g., visual clarity, contrast sensitivity, depth perception) remained unchanged, indicating that improvements were domain-specific rather than global. Overall, TVT demonstrated acceptability with selective improvements in visual search function and vision-related quality of life. Larger randomised

**Data availability statement:** The current article is accompanied by associated data and materials generated during the study and are available in the Open Science Framework at https://tinyurl.com/strokevision. Ethics approval, participant permissions, and all other relevant approvals were granted for this data sharing.

**Funding:** SD received funding from Northumbria University's internal Seedcorn Fund. The funder had no role in the study design, data collection and analysis, decision to publish, or preparation of this manuscript. As the funds were internal there is no associated grant number or URL for this fund.

**Competing interests:** The authors have declared that no competing interests exist.

controlled trials are needed to determine efficacy and comparative effectiveness against standard rehabilitation approaches.

## Author summary

Many stroke survivors experience visual field loss, which can make daily activities like reading and maintaining independence more challenging. We explored whether technological visuo-cognitive training (TVT) could be a feasible rehabilitation tool for people with post-stroke visual impairment. Using the Senaptec Sensory Station, 10 stroke survivors took part in a structured training program over two weeks. We assessed their visual cognition, usability of the system, and overall acceptability of the training before and after the intervention. Visual search performance improved, and participants reported significantly fewer visual difficulties and reduced fatigue. Importantly, they transferred improved scanning strategies to real-world activities, becoming more aware of their blind side when painting, crossing roads, and watching television. All participants became more confident with the technology and would recommend the intervention to others. However, some objective vision measures (clarity, contrast sensitivity) did not improve significantly. Barriers emerged including the need for adjustable difficulty levels and challenges with physical demands like standing and reaching. Our findings suggest that TVT has potential as a rehabilitation approach for stroke survivors with visual impairments, but further research is needed to determine its effectiveness compared to standard rehabilitation approaches and identify the best timing for delivering such training after stroke.

## Introduction

Post-stroke visual field loss is a common impairment experienced after a stroke, affecting between 20%-60% of all stroke survivors [1–3]. Homonymous hemianopia (HH), partial blindness in the visual fields to the right or left side of both eyes, is one such prominent visual impairment experienced post-stroke. Visual field loss can lead to impaired functioning and difficulty accomplishing everyday tasks independently, such as reading and navigating unfamiliar environments [4–6]. Consequently, survivors can become isolated, experience low mood and are often unable to work. These negative mental health effects can also extend to family and loved ones [7,8]. Visual field loss following stroke significantly reduces independence and quality of life [9,10]. Despite this, visual impairments such as HH are often overlooked, with stroke survivors identifying visual impairments as 'invisible', reflecting their experiences as an unknown and difficult symptom of stroke [11].

Visual rehabilitation interventions are typically delivered using the following approaches: restoring the visual field (restitution therapy), changing behaviour to compensate for lost visual function (compensation training) and substituting for the

visual field defect by using a device or extraneous modification (substitution therapy) [12]. Saccadic compensatory training (SCT) is one such compensation approach and one of the most promising rehabilitation options for stroke-related visual loss [13], significantly improving vision-related quality of life among patients [14–16], and recommended in the updated National Clinical Guideline for Stroke [17]. SCT tasks encourage compensation through making larger, more overt eye movements around the visual field. Previous research shows that rehabilitation using these types of scanning training can, alongside improved compensatory behaviour, lead to neural changes such as increased activity in the contralesional extrastriate cortex, suggesting that training of exploratory eye movements induces changes in the cortical representation of hemifields [18]. However, the evidence regarding efficacy of compensatory training for improving vision-related outcomes is limited and often low-quality [12,19].

Given the ongoing pressures on healthcare resources and the need for accessible long-term rehabilitation options after discharge from acute settings there is increasing interest in digital and home-based training tools. Technological visuo-cognitive training (TVT) programmes using technological devices (online, mobile or computer applications) have shown promise in neurological conditions such as Parkinson's disease [20–24]. However, their application to post-stroke visual field loss remains underexplored [25]. The potential mechanisms for benefit include promoting compensatory eye movements, enhancing attentional control, and supporting visuo-cognitive skills through repeated, gamified practice. A growing body of research has explored some digital and gamified interventions. For example, the Eye-Search study demonstrated that structured visual search training can yield meaningful improvements in both objective performance and patient-reported outcomes in adults with hemianopia, including those with co-existing neglect [26]. Additionally, in paediatric populations, children and young people with homonymous visual field loss have been found to be highly engaged with gamified training games, which also improved visual outcomes [27,28]. Together, these studies highlight the potential of digital and gamified training to enhance engagement and demonstrate feasibility across different populations, but evidence on their long-term effectiveness and real-world impact in stroke-related visual rehabilitation remains limited, underscoring the need for further trials using robust outcome measures [25].

One such option to deliver this type of intervention already exists: a multimodal TVT platform consisting of a dual-screen, touch-based interface designed to measure, train, and evaluate visual, cognitive, and visuomotor skills [29]. The Senaptec Sensory Station (SSS) is an interactive touch screen device consisting of a computer controlling two LCD monitors on a height adjustable pedestal. This tool has previously been effective in improving cognitive performance after concussion [30,31] and more recently as a rehabilitation tool in Parkinson's Disease [18]. Its gamified format is well-suited to encourage engagement and promote scanning behaviours, but its potential in stroke-related visual field loss is yet to be explored.

This pilot study aimed to explore the potential role of the SSS in stroke rehabilitation. Specifically, it sought to (1) examine the potential effects of TVT on stroke survivors with visual loss, and (2) explore the usability and accessibility of this technology and identify participant perspectives on its potential for home-based rehabilitation. By doing so, this study contributes to the growing literature on digital rehabilitation for hemianopia and addresses a gap regarding the acceptability of applying a visuo-cognitive training tool, already used in other neurological contexts, to stroke survivors in their own homes.

## Materials and methods

### Ethical approval

Ethical approval was provided by the Northumbria University Psychology Department Ethics Committee (Ref: 44683).

### Design

A single-arm pre-test, post-test design pilot study was conducted to investigate whether TVT is feasible for stroke survivors with visual loss and to explore the usability and accessibility of the Senaptec system as a rehabilitation tool, including

baseline assessment, SSS training and post-assessment. While the psychometric properties of the SSS have only been partially explored, with one study in healthy college-aged individuals reporting good reliability for tasks such as go/no-go, multiple-object tracking, eye-hand coordination, depth perception, and reaction time, but poorer reliability for other visual skills [32], these measures have not yet been validated in clinical populations. As this was a pilot study, our primary aim was to assess usability and acceptability, with the evaluation of psychometric properties identified as an important future step.

## Participants

10 stroke survivors were recruited for this project from an existing database. All participants had been diagnosed with a non-progressive left or right visual field defect, demonstrated sufficient cognitive ability as determined by clinical judgment and were able to give informed consent. The minimum time since homonymous visual field defect onset was three months. Those with comorbid ophthalmic field defects or oculomotor problems were excluded. Full participant demographics can be found in Table 1.

## Clinical assessments

All assessments and training were completed at Northumbria University, Coach Lane Campus. Assessments and training sessions were completed across a two-week period (see Fig 1). Assessments lasted ~1.5 hours and training sessions lasted ~30 minutes. Participants completed a baseline assessment (Day 0), followed by five training sessions on alternate days (Days 1–11), and a final follow-up assessment on Day 12. The timing of the follow-up was fixed, with no variation across participants. At the initial baseline assessment, demographic data were collected, including age, gender, type and time since stroke, and participants were asked to self-report any ongoing impairments they experienced (e.g., vision, speech, motor, memory, cognitive, pain). Participants completed a battery of computer-based visuo-cognitive measures on the SSS, paper-based assessments and subjective measures assessing quality of life, activities of daily living and visual behaviour. Assessment tasks were counterbalanced across participants to minimise potential order effects.

The Senaptec assessment covered ten sensory parameters completed on the SSS (see Figs 2–4); visual clarity, depth perception, near-far quickness, perception span, multiple object tracking, reaction time, target capture, hand-eye coordination, and go/no go (see Table 2 for a breakdown of these tasks). Assessments were carried out using a combination of the large screen, companion tablet and mobile phone on the SSS. Participants were required to stand or sit close

**Table 1. Participant demographics.**

| ID | Gender | Age | Type of stroke | Time since stroke | Self-reported Impairment* | | | | | |
|----|--------|-----|----------------|-------------------|------|--------|-------|--------|-----------|------|
| | | | | | Vision | Speech | Motor | Memory | Cognitive | Pain |
| 01 | Male | 73 | Ischaemic | 17 years | Yes | No | No | Yes | No | No |
| 02 | Male | 61 | Haemorrhagic | 13 years | Yes | No | Yes | Yes | Yes | Yes |
| 03 | Male | 69 | Ischaemic | 18 years | Yes | No | Yes | No | No | No |
| 04 | Female | 78 | Ischaemic | 1 year | Yes | No | No | Yes | Yes | Yes |
| 05 | Male | 50 | Ischaemic | 4 years | Yes | No | Yes | Yes | No | No |
| 06 | Male | 64 | Haemorrhagic | 12 years | Yes | No | Yes | Yes | No | No |
| 07 | Male | 79 | Not known | 2 years | Yes | No | Yes | Yes | Yes | No |
| 08 | Male | 62 | Not known | 15 years | Yes | No | Yes | Yes | Yes | No |
| 09 | Female | 43 | Haemorrhagic | 7 years | Yes | No | Yes | No | Yes | Yes |
| 10 | Male | 71 | Ischaemic | 30 years | Yes | Yes | Yes | Yes | Yes | Yes |

*Participant demographics and self-reported stroke-related impairments (vision, speech, motor, memory, cognitive, pain) collected during the baseline demographic and visual health assessment.

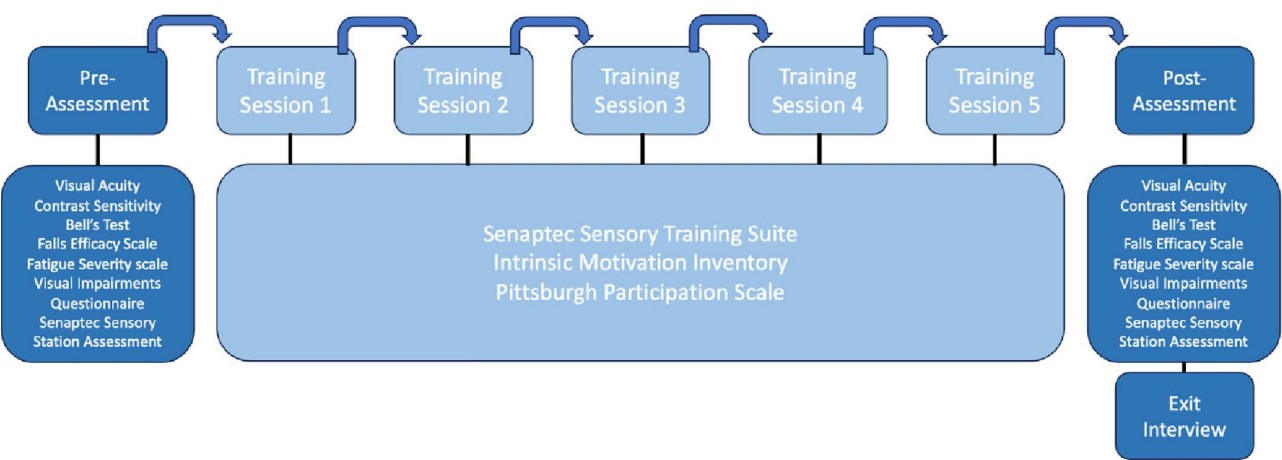

**Fig 1. Flowchart of assessment and training protocol.**

or far from the screen depending on the task and participant mobility and preference. Throughout the assessment, the researcher gave instructions to the participant, and each task consisted of a demonstration, practice run and test phase. Once participants completed a task they would progress to the next. Some tasks could not be completed due to participant mobility issues. Where this was the case, the researcher progressed the participant to the next task and made a note of these usability issues.

After completing the SSS assessment, participants completed a series of non-computerised measures. These included standardised visual tests (the Logarithm of the Minimum Angle of Resolution (LogMAR) and Mars Contrast Sensitivity tests), self-report questionnaires (the Falls efficacy scale (FES-I; [33]) to assess any changes in the level of concern surrounding falling in social and physical activities inside or outside the home, the Fatigue Severity Scale (FSS; [34]) to gauge participants perceptions of their fatigue over the course of the study, a Daily Activities questionnaire exploring difficulty experienced in activities of daily living, the Visual Impairments Questionnaire (VIQ; adapted from Kerkhoff et al. [35]) to investigate participants confidence in their visual ability and the Bells test [36] a cancellation task with distractors requiring visual exploration. Participants were also asked to rate the usability of the system using the Systems Usability Scale [37].

Assessments were identical at both timepoints, with the inclusion of a short interview in the second assessment to explore participants perspectives of the system, visual training, and the use of technology in their rehabilitation. Interviews lasted up to 30 minutes and were transcribed verbatim.

## Technological visuo-cognitive training

Participants completed 5 training sessions over the course of a 10-day period. Training sessions lasted an average of 30 minutes, including both the SSS training and paper-based assessments. At the start of the intervention all participants were given a demonstration of each task by the researcher and a short practice. In each training session participants completed ten sensory training modules (see Figs 5–7): Eye-Hand Coordination, Go No Go, Perception Training, Response Inhibition, Spatial Sequence, Spatial Memory, Multiple Object Tracking, Visual Search, Shape Cancellation and Visual-Motor Integration. The order of tasks was the same for all participants across sessions. The progression from one module to another was manually done by the researcher, with opportunity for breaks in between modules if needed. All participants were able to complete all modules across all training sessions.

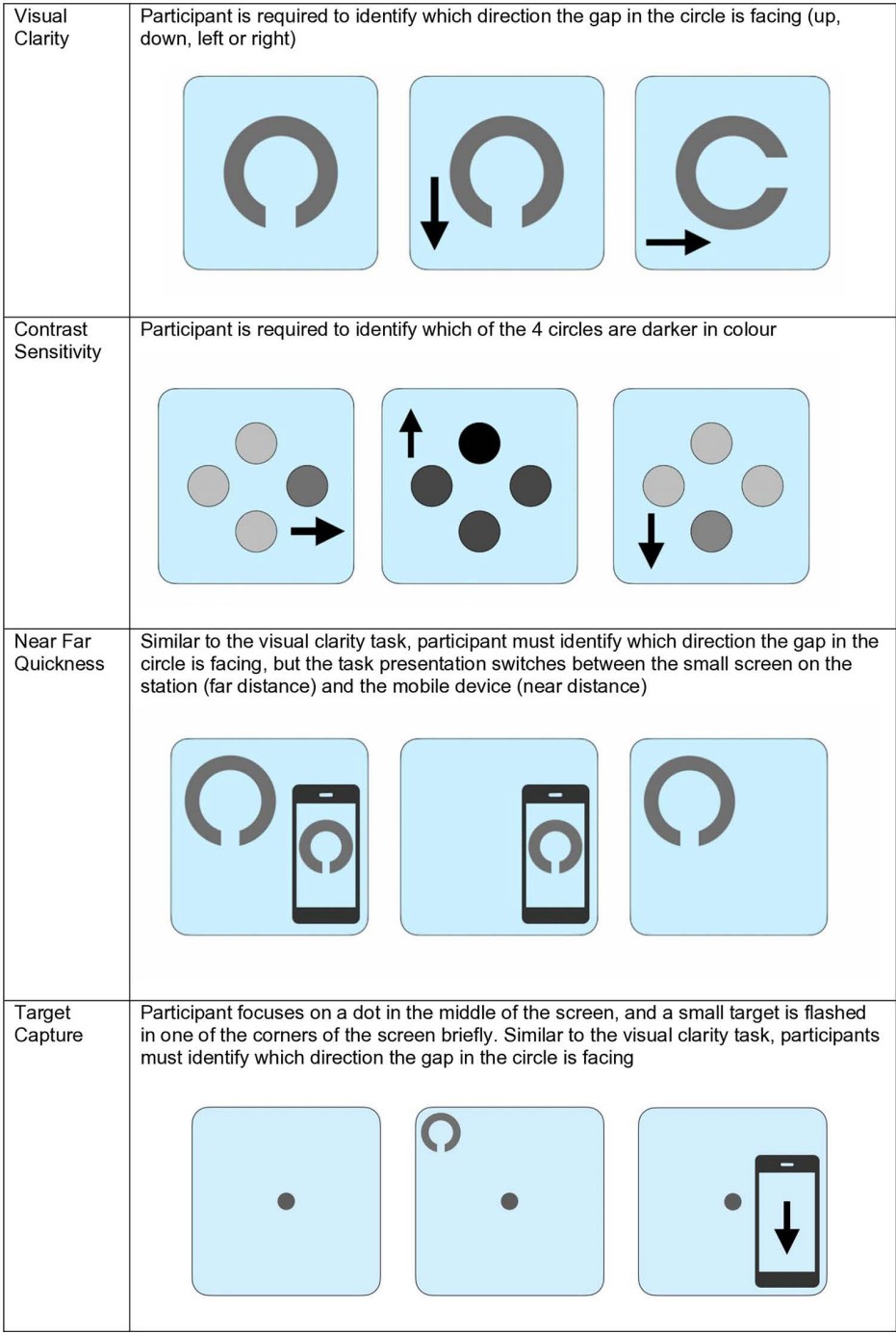

| Visual Clarity | Participant is required to identify which direction the gap in the circle is facing (up, down, left or right) |
| Contrast Sensitivity | Participant is required to identify which of the 4 circles are darker in colour |
| Near Far Quickness | Similar to the visual clarity task, participant must identify which direction the gap in the circle is facing, but the task presentation switches between the small screen on the station (far distance) and the mobile device (near distance) |
| Target Capture | Participant focuses on a dot in the middle of the screen, and a small target is flashed in one of the corners of the screen briefly. Similar to the visual clarity task, participants must identify which direction the gap in the circle is facing |

**Fig 2. Senaptec assessment tasks: visual clarity, contrast sensitivity, near-far quickness and target capture.** Assessment tasks measuring visual clarity at left, right, and binocular viewing conditions; depth perception in forward, left, and right directions; and near-far quickness performance.

The Intrinsic Motivation Inventory (IMI; [38]) was administered after each training session to assess participants' motivation and engagement with the technology. While the IMI was not designed as an acceptability measure, its sub-scales provide proxy indicators of key acceptability constructs [39]: interest/enjoyment reflects affective attitude toward

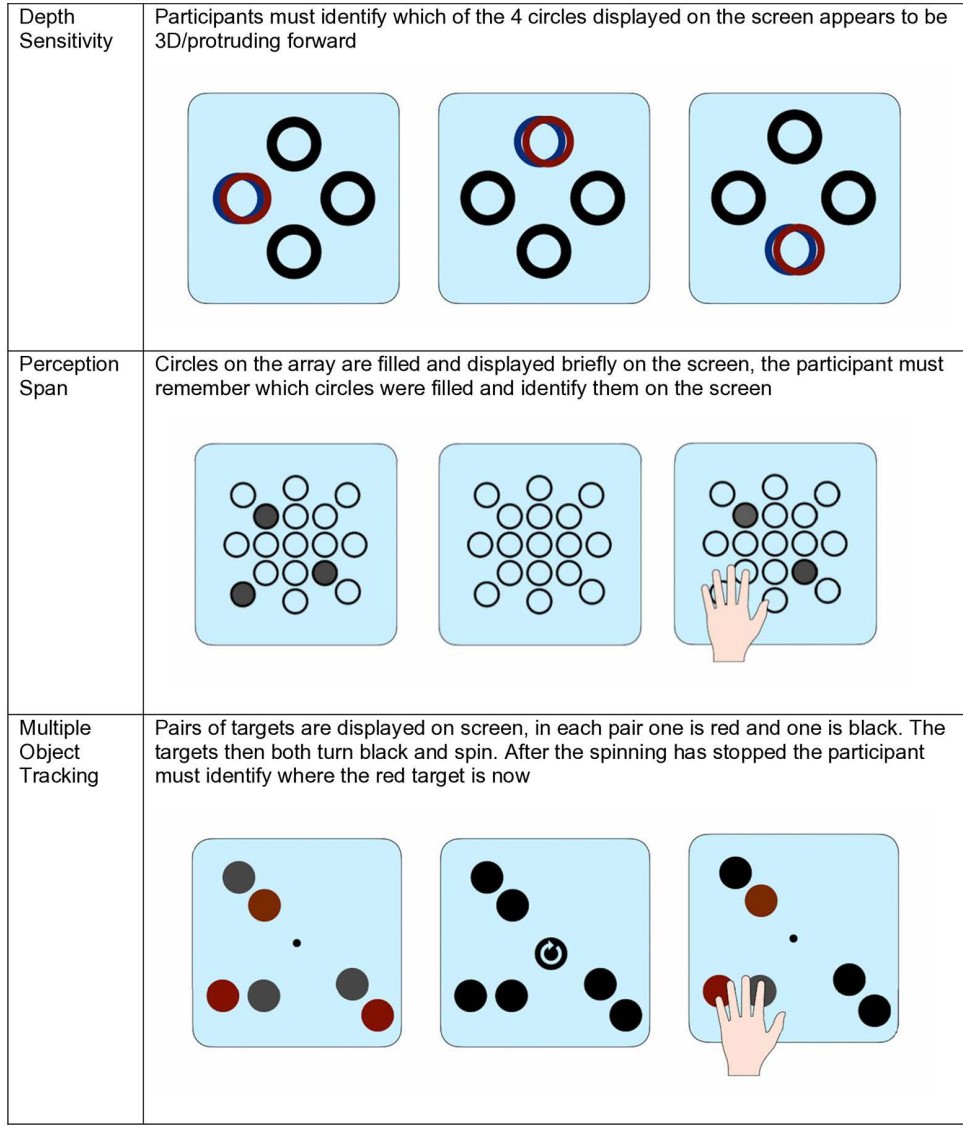

| Depth Sensitivity | Participants must identify which of the 4 circles displayed on the screen appears to be 3D/protruding forward |
| --- | --- |
| Perception Span | Circles on the array are filled and displayed briefly on the screen, the participant must remember which circles were filled and identify them on the screen |
| Multiple Object Tracking | Pairs of targets are displayed on screen, in each pair one is red and one is black. The targets then both turn black and spin. After the spinning has stopped the participant must identify where the red target is now |

**Fig 3. Senaptec assessment tasks: depth sensitivity, perception span, and multiple object tracking.** Assessment tasks measuring depth sensitivity; perception span (visual field awareness); and multiple object tracking.

the intervention, perceived competence indicates self-efficacy, and value/usefulness captures perceived effectiveness. These were supplemented by qualitative interviews to provide a comprehensive assessment of acceptability. Finally, the researcher rated participant participation and engagement in the sessions using the Pittsburgh Participation Scale [40]. Throughout the training sessions, written observations were also made to record any significant comments or actions from participants.

## Statistical analysis

Alpha level was set at $p = 0.05$, with no control for multiple comparisons due to the pilot nature of the study.

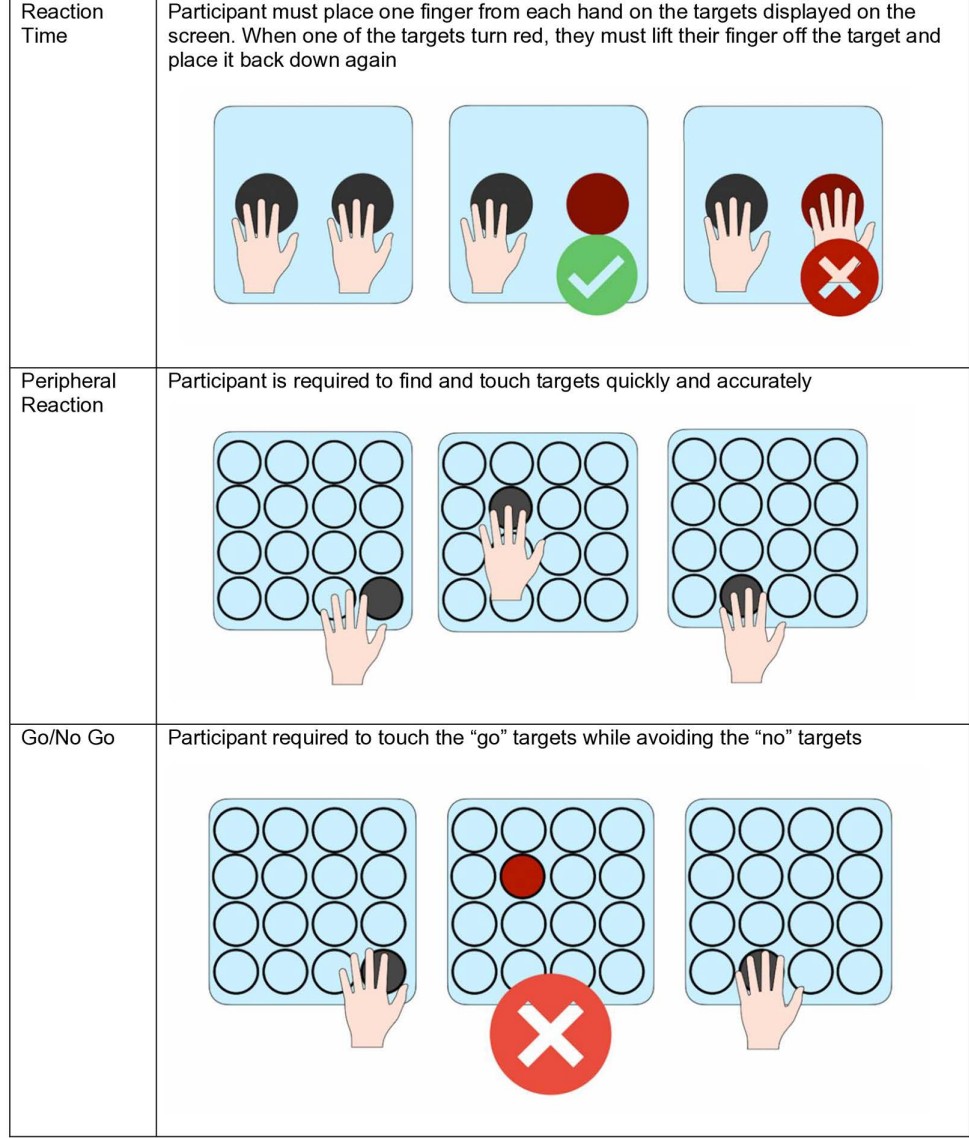

**Fig 4. Senaptec assessment tasks: reaction time, peripheral reaction, and go/no-go tasks.** Assessment tasks measuring reaction time; peripheral reaction (eye-hand coordination); and go/no-go discrimination performance.

To examine whether TVT is useful in stroke survivors, a series of repeated measures t-tests were used to analyse computerised visuo-cognitive data, comparing pre- and post-assessment. Additionally, we explored whether stroke survivors improved in their TVT performance over time via repeated measures ANOVAs across sessions 1–5. Greenhouse-Geisser corrections are reported where Mauchly's test of Sphericity was significant, indicating a violation of the sphericity assumption.

To explore the usability and accessibility of the TVT, t-tests were used to compare the pre and post systems usability scale and Intrinsic Motivation Inventory.

**Table 2. Pre-Post assessment comparison for senaptec sensory station modules.**

| Module | Pre-Assessment M (SD) | Post-Assessment M (SD) | T values | DF | 95% CI | p | d |
|---|---|---|---|---|---|---|---|
| *Visual Clarity* | | | | | | | |
| Left | 0.228 (0.265) | 0.321 (0.240) | -1.074 | 9 | [-0.287, 0.102] | .311 | -0.340 |
| Right | 0.389 (0.266) | 0.371 (0.392) | 0.242 | 9 | [-0.148, 0.183] | .814 | 0.076 |
| Both | 0.290 (0.176) | 0.149 (0.242) | 1.781 | 9 | [-0.038, 0.319] | .109 | 0.563 |
| *Contrast Sensitivity* | | | | | | | |
| 6 | 1.560 (0.372) | 1.610 (0.393) | -0.275 | 9 | [-0.461, 0.361] | .789 | -0.087 |
| 18 | 0.644 (0.207) | 0.800 (0.430) | -1.224 | 8 | [-0.449, 0.137] | .256 | -0.408 |
| *Near Far Quickness* | 7.400 (3.502) | 6.100 (3.381) | 2.112 | 9 | [-0.092, 2.692] | .064 | 0.668 |
| *Target Capture* | 452.50 (90.10) | 402.50 (109.58) | 2.301 | 9 | [0.842, 99.158] | .047* | 0.728 |
| *Depth Perception* | | | | | | | |
| Forward | 226.10 (36.60) | 235.40 (9.70) | -0.938 | 9 | [-31.722, 13.122] | .373 | -0.297 |
| Left | 202.70 (66.98) | 235.40 (9.70) | -1.546 | 9 | [-80.562, 15.162] | .157 | -0.489 |
| Right | 220.00 (50.00) | 215.80 (57.66) | 0.194 | 9 | [-44.671, 53.071] | .850 | 0.061 |
| *Perception Span* | 11.00 (10.97) | 13.30 (11.57) | -0.637 | 9 | [-10.464, 5.864] | .540 | -0.202 |
| *Multiple Object Tracking* | 0.475 (0.210) | 0.515 (0.214) | -0.733 | 9 | [-0.163, 0.083] | .482 | -0.232 |
| *Reaction Time* | 493.50 (116.28) | 462.90 (60.89) | 1.289 | 9 | [-23.102, 84.302] | .230 | 0.408 |
| *Peripheral Reaction* | 1474.74 (826.68) | 1325.88 (670.00) | 2.090 | 9 | [-12.228, 309.953] | .066 | 0.661 |

Note. *p <.05. CI = Confidence Interval; d = Cohen's d effect size. Mean values are reported with Standard Deviations: M (SD).

## Qualitative analysis

Interviews were transcribed verbatim and checked against the audio recordings. Data were analysed using Template Analysis [41]. An initial template was developed based on the study's research questions, covering: (1) acceptability and confidence with technology, (2) enjoyment and engagement, and (3) perceived changes in visual behaviour. SD led the analysis with input from AF. Following familiarisation with transcripts through repeated reading and involvement in data collection, a preliminary coding template was developed from three transcripts. Both analysts independently applied the template to all ten transcripts, with flexibility to add, modify, or remove codes as needed. Regular consensus meetings were held to refine code definitions and resolve interpretive differences. The template was iteratively revised throughout analysis, with coding progressing from descriptive to more interpretive levels. Final themes were identified through synthesis of coded data.

## Researcher positionality

SD is a psychologist and mixed-methods researcher with 15 years' experience working with stroke survivors and led the design and conduct of the study. AF is a PhD researcher exploring gamified rehabilitation for stroke survivors with visual loss and worked as a research assistant on this project. Their combined expertise informed familiarity with the topic, while potential assumptions were addressed through ongoing reflection and collaborative discussion throughout the analysis.

## Results

The current article is accompanied by associated data and materials generated during the study and are available in the Open Science Framework at https://osf.io/pwvde. Ethics approval, participant permissions, and all other relevant approvals were granted for this data sharing.

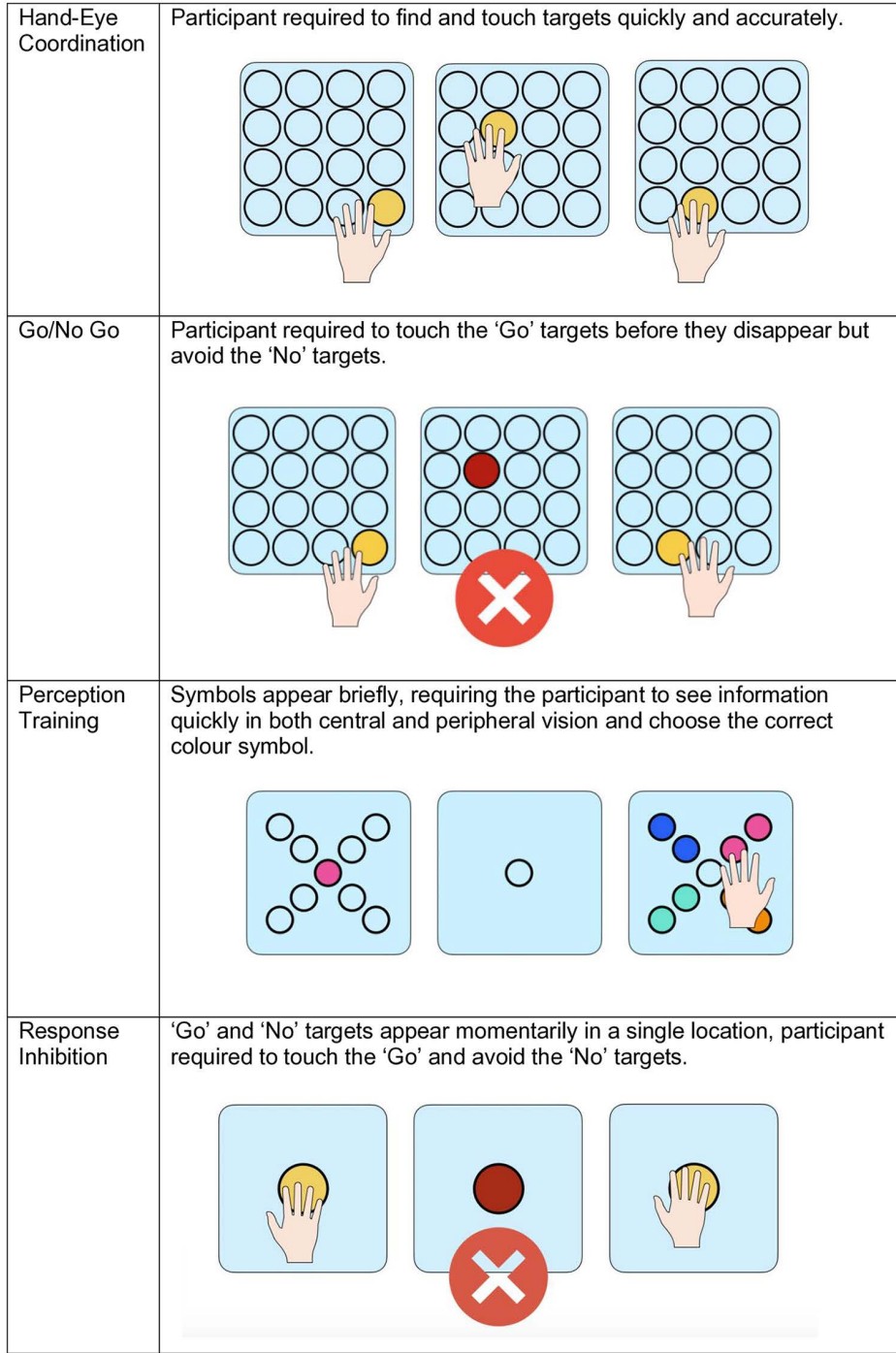

| Hand-Eye Coordination | Participant required to find and touch targets quickly and accurately. |
| Go/No Go | Participant required to touch the 'Go' targets before they disappear but avoid the 'No' targets. |
| Perception Training | Symbols appear briefly, requiring the participant to see information quickly in both central and peripheral vision and choose the correct colour symbol. |
| Response Inhibition | 'Go' and 'No' targets appear momentarily in a single location, participant required to touch the 'Go' and avoid the 'No' targets. |

**Fig 5. Senaptec training modules: hand-eye coordination, go/no-go, perception training, and response inhibition.** Gamified training tasks for hand-eye coordination, go/no-go discrimination, perception training, and response inhibition.

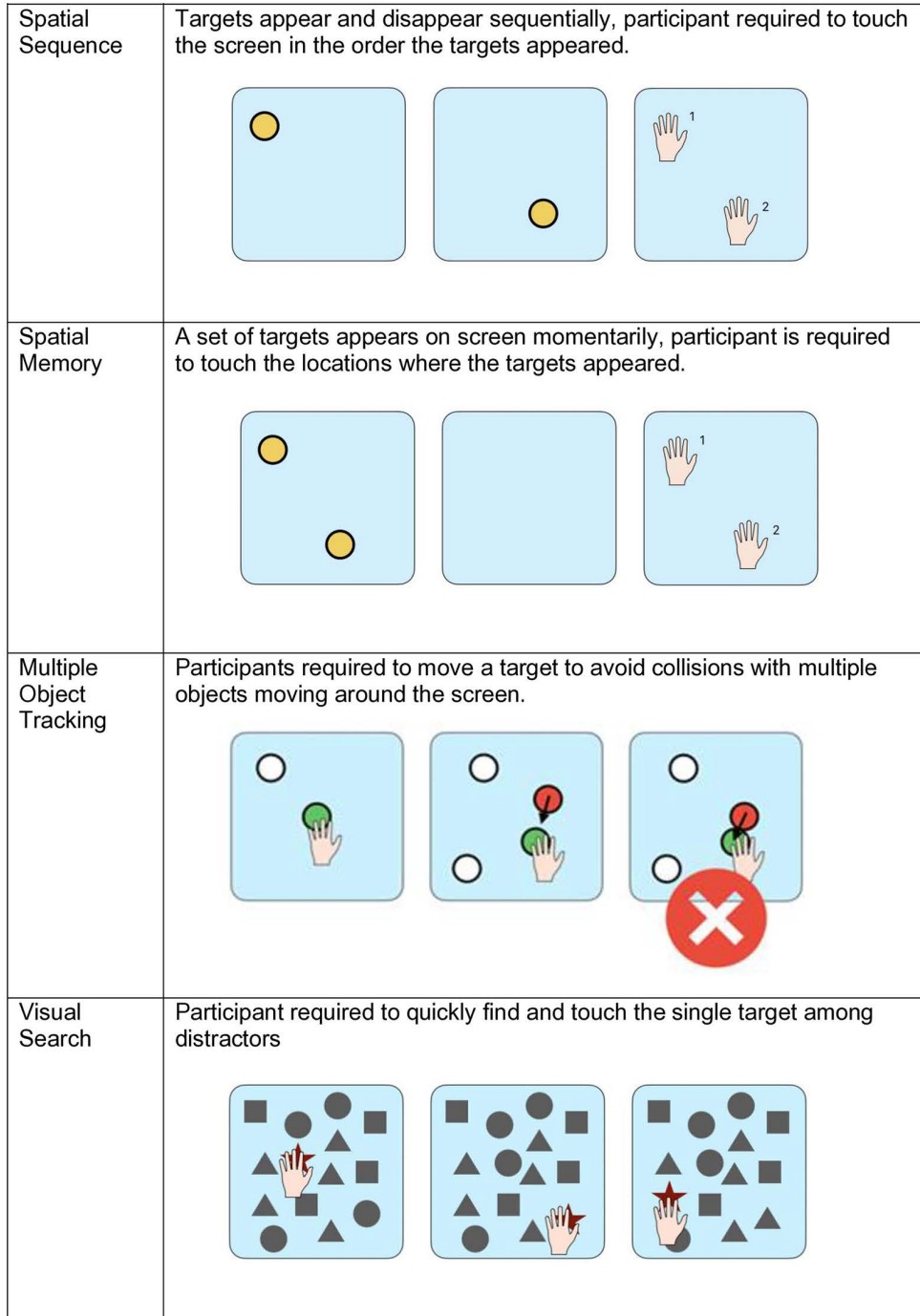

| Spatial Sequence | Targets appear and disappear sequentially, participant required to touch the screen in the order the targets appeared. |
| Spatial Memory | A set of targets appears on screen momentarily, participant is required to touch the locations where the targets appeared. |
| Multiple Object Tracking | Participants required to move a target to avoid collisions with multiple objects moving around the screen. |
| Visual Search | Participant required to quickly find and touch the single target among distractors |

**Fig 6. Senaptec training modules: spatial sequence, spatial memory, multiple object tracking, and visual search.** Gamified training tasks for spatial sequence learning, spatial memory, multiple object tracking, and visual search.

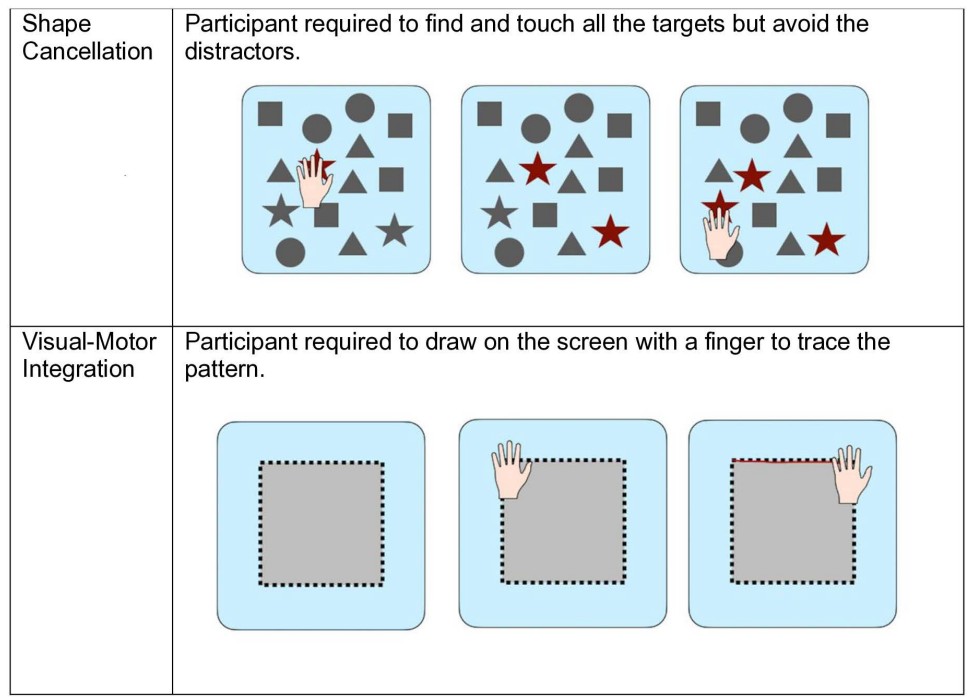

**Fig 7. Senaptec training modules: shape cancellation and visuo-motor integration.** Gamified training tasks for shape cancellation (visual search with distractors) and visuo-motor integration (drawing patterns).

### Technological visuo-cognitive assessment

Paired t-tests examined pre- to post-assessment changes on SSS measures (see Table 2). A significant improvement was observed for target capture, with participants demonstrating faster response times post-assessment (Mpre = 452.50, SDpre = 90.10; Mpost = 402.50, SDpost = 109.58), t(9) = 2.301, p = .047, d = .728. No significant changes were observed for visual clarity (left: t(9) = -1.074, p = .311, d = -.340; right: t(9) = 0.242, p = .814, d = .076; both eyes: t(9) = 1.781, p = .109, d = .563), contrast sensitivity (6 cpd: t(9) = -0.275, p = .789, d = -.087; 18 cpd: t(8) = -1.224, p = .256, d = -.408), depth perception (forward: t(9) = -0.938, p = .373, d = -.297; left: t(9) = -1.546, p = .157, d = -.489; right: t(9) = 0.194, p = .850, d = .061), near-far quickness (t(9) = 2.112, p = .064, d = .668), perception span (t(9) = -0.637, p = .540, d = -.202), multiple object tracking proportion correct (t(9) = -0.733, p = .482, d = -.232), eye-hand coordination reaction time (t(9) = 2.090, p = .066, d = .661), go/no-go total score (no variance observed at either timepoint), or reaction time (t(9) = 1.289, p = .230, d = .408).

### Standardised clinical vision assessments

Significant improvements were observed in Bell's Test performance, with participants finding more targets post-assessment (Mpre = 24.00, SDpre = 8.05; Mpost = 30.25, SDpost = 3.20), t(7) = -2.617, p = .035, d = -.925. LogMAR binocular visual acuity showed a trend toward improvement approaching significance (Mpre = 0.09, SDpre = 0.20; Mpost = 0.03, SDpost = 0.18), t(9) = 2.250, p = .051, d = .712. No significant change was observed for Mars contrast sensitivity, t(9) = 0.737, p = .480, d = .233.

### Patient-reported outcomes

Participants reported significant reductions in visual difficulties (VIQ) following the intervention (Mpre = 2.28, SDpre = 0.91; Mpost = 1.92, SDpost = 0.95), t(9) = 2.903, p = .018, d = .918. Fatigue severity decreased significantly

(Mpre = 53.50, SDpre = 12.35; Mpost = 50.40, SDpost = 12.54), t(9) = 2.514, p = .033, d = .795. Fear of falling showed a non-significant trend toward reduced concern (Mpre = 40.80, SDpre = 15.94; Mpost = 35.80, SDpost = 11.64), t(9) = 1.450, p = .181, d = .459.

## Technological visuo-cognitive training

Repeated-measures ANOVAs were conducted to examine whether performance improved across the five training sessions for each training module. Descriptive statistics and F-statistics for all modules are presented in Table 3. Mauchly's test was used to assess sphericity. Where the assumption was violated (p < .05), Greenhouse-Geisser corrected values are reported.

Significant improvements across training sessions were observed for two modules. Hand-eye coordination showed a significant effect of session, $F(4, 36) = 4.700$, p = .004, $\eta p^2 = .343$, with participants finding more targets in later sessions (Session 1: M = 21.40, SD = 5.70; Session 5: M = 25.10, SD = 6.26). Visuomotor integration accuracy also improved significantly across sessions, $F(4, 36) = 3.569$, p = .039, $\eta p^2 = .284$, with performance increasing from Session 1 (M = 85.90, SD = 19.47) to Session 5 (M = 101.00, SD = 11.76).

No significant effects of training session were found for the remaining eight modules: Go/No Go, $F(4, 36) = 1.826$, p = .145, $\eta p^2 = .169$; Perception Training, $F(1.99, 17.87) = 0.236$, p = .791, $\eta p^2 = .026$; Response Inhibition, $F(4, 36) = 0.683$, p = .609, $\eta p^2 = .071$; Spatial Sequence, $F(4, 36) = 1.924$, p = .128, $\eta p^2 = .176$; Spatial Memory, $F(2.25, 20.27) = 1.289$, p = .300, $\eta p^2 = .125$; Multiple Object Tracking, $F(4, 36) = 0.707$, p = .592, $\eta p^2 = .073$; Visual Search, $F(1.87, 16.84) = 0.550$, p = .576, $\eta p^2 = .058$; and Shape Cancellation, $F(4, 36) = 0.536$, p = .710, $\eta p^2 = .056$.

## Acceptability, usability and feasibility

**Acceptability.** Repeated-measures ANOVAs examined Intrinsic Motivation Inventory subscales across training sessions. Perceived competence increased significantly, $F(4, 36) = 4.117$, p = .008, $\eta p^2 = .314$ (Session 1: M = 25.20,

Table 3. Performance on visuo-cognitive training modules across five training sessions.

| Module | Session 1 M (SD) | Session 2 M (SD) | Session 3 M (SD) | Session 4 M (SD) | Session 5 M (SD) | F | df | p | ηp² |
|---|---|---|---|---|---|---|---|---|---|
| *Hand-Eye Coordination* | 21.40 (5.70) | 24.30 (6.08) | 24.90 (7.82) | 24.50 (6.57) | 25.10 (6.26) | 4.700 | 4, 36 | .004* | .343 |
| *Go/No Go* | 9.80 (10.60) | 12.20 (9.24) | 11.30 (7.95) | 13.20 (8.27) | 14.40 (8.55) | 1.826 | 4, 36 | .145 | .169 |
| *Perception Training* | 11184.70 (16258.61) | 13768.90 (16610.86) | 13718.10 (17303.26) | 13864.60 (15518.05) | 13272.80 (15685.11) | 0.236 | 1.99, 17.87 | .791 | .026 |
| *Response Inhibition* | 8.20 (2.20) | 9.00 (1.49) | 8.90 (2.81) | 9.20 (1.03) | 9.40 (1.58) | 0.683 | 4, 36 | .609 | .071 |
| *Spatial Sequence* | -557.00 (2579.68) | -1472.70 (2738.81) | 1003.00 (2092.89) | -238.40 (3675.76) | 336.70 (2457.05) | 1.924 | 4, 36 | .128 | .176 |
| *Spatial Memory* | 1496.90 (1777.74) | 1766.60 (2589.37) | 1871.70 (1519.32) | 2226.60 (1420.87) | 2047.40 (1646.44) | 1.289 | 2.25, 20.27 | .300 | .125 |
| *Multiple Object Tracking* | 501.40 (100.76) | 469.00 (152.79) | 448.90 (131.10) | 511.70 (132.91) | 512.60 (112.02) | 0.707 | 4, 36 | .592 | .073 |
| *Visual Search* | 42.46 (26.50) | 38.59 (21.96) | 40.27 (32.76) | 41.57 (23.75) | 37.51 (27.83) | 0.550 | 1.87, 16.84 | .576 | .058 |
| *Shape Cancellation* | 53.20 (6.39) | 53.80 (8.00) | 50.50 (8.25) | 52.80 (7.36) | 53.50 (8.34) | 0.536 | 4, 36 | .710 | .056 |
| *Visuomotor Integration* | 85.90 (19.47) | 93.40 (16.59) | 96.20 (7.22) | 97.70 (5.89) | 101.00 (11.76) | 3.569 | 2.40, 21.58 | .039* | .284 |

Note. *p < .05. ηp² = partial eta squared. Degrees of freedom adjusted using Greenhouse-Geisser correction where Mauchly's test indicated violation of sphericity (Perception Training, Spatial Memory, Visual Search, Visuomotor Integration).

SD = 9.72; Session 5: M = 31.10, SD = 7.92), indicating participants developed greater confidence with the technology over time. No significant changes were observed for interest/enjoyment, $F(2.08, 18.75) = 0.748$, $p = .492$, $\eta p^2 = .077$; effort, $F(2.47, 22.22) = 1.582$, $p = .225$, $\eta p^2 = .150$; pressure, $F(1.80, 16.21) = 2.130$, $p = .154$, $\eta p^2 = .191$; or value, $F(1.59, 14.30) = 3.512$, $p = .066$, $\eta p^2 = .281$.

**Usability.** Post-intervention SUS scores (M = 84.25, SD = 10.34, range = 65–100) indicated excellent usability, substantially exceeding the industry average of 68 [42].

**Feasibility.** Although the intervention and training sessions therein were not particularly intensive, that all ten participants completed the full study protocol with no withdrawals indicates at least suitable feasibility. Training sessions were completed within the anticipated 30-minute timeframe. While it is important to note that participant mobility limitations posed a barrier to assessment tasks, particularly balance difficulties affecting standing tasks and reaching touchscreen areas, these did not prevent training protocol completion or affect session attendance.

## Template analysis

Analysis of the interview data identified three key themes: (1) enjoyment of using the technology, (2) barriers to use, recommendations and system design, and (3) impacts of training (positive and negative). These themes provide insight into participants' experiences of technological visuo-cognitive training and its feasibility as a rehabilitation tool for stroke survivors with visual impairments.

**Theme 1: Enjoyment of using the technology.** Participants generally reported positive experiences with the training, with enjoyment stemming from multiple sources. The gamified nature of the tasks was particularly valued, with participants appreciating both the cognitive engagement and the sense of purpose the activities provided. One participant distinguished the therapeutic purpose from entertainment:

> *"They have a purpose. I would never play them for entertainment, but to play them for a test is different, it's a different way of thinking about things."* (PB01)

The cognitive challenge was frequently mentioned as a positive aspect. Participants valued the mental stimulation, with several describing it as "brain training." As one participant explained:

> *"The jolt they give your brain is an advantage. So it's like using your brain in slightly unexpected ways. Use it or lose it, don't you?...It's a different dimension to just cooking and washing and stuff, you're using your brain in a different way, different parts of the brain."* (MH02)

This cognitive engagement was seen as particularly important given the limited rehabilitation options available post-discharge.

**Theme 2: Barriers to use, recommendations and system design.** While participants were generally positive about the intervention, several practical barriers emerged that have important implications for real-world implementation. Participants emphasised the role of adaptive difficulty in these gamified tasks with some aspects of the tasks (e.g., stimulus speed, difficulty of task) mentioned as barriers to use, particularly for those with more severe visual field loss.

> *"All strokes are different. It might work for everybody else but me. I just need it to be slower. Because if it's too fast I don't see it, you're not going to learn anything."* (PB01)

Physical limitations also affected participation. Several participants noted difficulties with tasks requiring standing, bimanual coordination, or reaching across large screens. One participant described:

*"Just in terms of my lack of abilities, like reaching...reaching parts of the screen and the tablet, like I kind of stand up, but obviously I'm concentrating on standing up as well as concentrating on what's on the screen."* (SC09)

This participant suggested that using a tablet at home might reduce these physical demands compared to the large station screen used in the study.

The question of ecological validity was also raised:

*"If we're doing the research on a big screen is it kind of like researching rats compared to humans?"* (SC09)

This insightful observation highlighted the importance of considering real-world implementation, with participants suggesting modifications for home use including smaller screens, adjustable difficulty, and consideration of visual factors such as contrast and colour. As one participant noted:

*"I find certain colours stand out better than like white and stuff. See that red, that showed up nicely...definitely better colours, brighter colours on darker background are better."* (GS08)

Despite these barriers, participants were overwhelmingly willing to recommend the training to others. All ten participants indicated they would recommend the intervention, though several noted its potential benefits might be greater for those earlier in their stroke recovery.

*"Mainly probably beneficial to newer ones, so that they can sort of see what they can do...And again, for somebody who's maybe longer in the tooth, a few years from the date, it could open their eyes a bit, maybe not as much as a new one."* (MC03)

Most participants expressed desire to continue training if given the opportunity, though practical considerations such as frequency of sessions were mentioned. The willingness to continue appeared linked to perceived benefit and the value placed on any opportunity for improvement. As one participant explained:

*"I think because any improvement in any part of me, since my stroke, is important to see. Because you can get very miserable sometimes about things, not being able to do stuff, and feeling useless."* (SC09)

**Theme 3: Impacts of training (positive and negative).** The most striking positive impact reported was increased awareness of visual scanning strategies. Several participants described becoming more conscious of the need to look toward their blind side in daily activities. One participant provided concrete examples:

*"When I've been painting pictures I haven't always - I've remembered more to look at both sides of the paper. Whereas before I'd just paint on the right side...I think with things like crossing roads and stuff, I seem to remember more to look both ways rather than just the one...Even just like remembering to eat the food on both sides of the plate."* (SC09)

This transfer to real-world activities was echoed by another participant:

*"I'm more aware of the need to look, to look at my blind spot basically...I suppose to some extent watching TV's now a bit easier...I'm obviously aware I'm missing things on this side, but now I'm learning to look more to the right, so I'm picking them up."* (CL06)

Some participants reported perceived improvements in speed and reaction time:

*"I thought I was getting quicker. I seem to have been speeding up from where I first started off at quite a low level. Myself I feel a bit quicker."* (JW10)

*"I was a bit faster, because I was using both eyes at the same time rather than one."* (MH02)

However, it is important to note that not all participants reported noticeable changes. Several participants explicitly stated they had not noticed differences in their vision or daily functioning, though some acknowledged the difficulty in quantifying subtle changes. As one participant reflected:

*"It's hard to gauge, isn't it? You know you can't really put a number on it."* (MH02)

Memory and cognitive difficulties also affected some participants' ability to track changes over time, with one participant noting:

*"I don't remember doing it before. I have no idea. I can tell you how I feel now."* (PB01)

The time since stroke appeared to influence expectations of benefit, with those many years post-stroke expressing more realistic or limited expectations for personal improvement while still recognising potential value for others. One participant stated:

*"To be honest I don't think it will change my eyesight...I don't think I'm going to improve my situation type of thing. It might help a lot of other people."* (MC03)

Despite this pragmatism, these participants still valued the training for its cognitive stimulation and the opportunity for any incremental improvement.

The intervention did produce fatigue in some participants, though this was generally interpreted positively as evidence of meaningful cognitive engagement. One participant explained:

*"I think when I get home I find that my brain is tired. But that's good because it means I've been exercising it...I always feel as though it's done me a lot of good, exercising you know, exercising the brain."* (JB04)

This fatigue was distinguished from physical tiredness and seen as an indicator that the training was challenging their cognitive systems appropriately.

## Discussion

Our findings present a mixed but promising picture regarding the efficacy of the SSS intervention for use in stroke rehabilitation. Findings revealed selective improvements in visuo-cognitive function, significant benefits in patient-reported outcomes, and increased technology competence. This pilot study highlights TVT as a promising tool for post-stroke visual impairment, particularly for fostering adaptive visual scanning strategies and improving vision-related quality of life [9].

### Visuo-cognitive improvement with TVT in stroke survivors

Significant improvements were observed on target capture and Bell's test measures, suggesting participants adopted more optimal search strategies. Importantly, qualitative data revealed that participants transferred these improvements to daily activities, reporting increased visual scanning awareness when painting, crossing roads, eating, and watching television. This functional transfer is significant, as previous visual search training has been limited by task-specificity, where

improvements did not generalise to other activities such as reading [42,43]. The gamified, multi-modal nature of the SSS may facilitate broader generalisation of learned strategies, addressing a known limitation in compensatory visual training [35].

Several objective measures, however, showed no significant improvements: visual clarity, contrast sensitivity, depth perception, perception span, multiple object tracking, and various other visuo-cognitive modules. This selective pattern suggests the intervention effectively enhanced visual search and exploration but may not produce uniform improvements across all domains. It is possible that the training tasks were not adequately tailored to address these specific skills, or that the duration and intensity of the intervention were insufficient to elicit identifiable change [12]. Measures approaching significance (e.g., near-far quickness and eye-hand coordination reaction time) may demonstrate benefits in larger samples.

### Patient-reported outcomes

Participants reported significant reductions in visual difficulties with a strong effect size, indicating meaningful improvements in vision-related quality of life. This finding is particularly important as subjective visual difficulties often have greater impact on daily functioning and participation than objective visual measures alone [8,15]. Fatigue severity also decreased significantly. The quantitative reduction reflects the intervention's efficiency; the SSS requires approximately 30 minutes per session, substantially less time-consuming than traditional compensatory training techniques which typically involve one hour of daily training for at least one month [44]. This efficiency is particularly important as fatigue is a major barrier to rehabilitation engagement in stroke populations [45,46].

Qualitatively, participants distinguished cognitive from physical fatigue, interpreting cognitive fatigue as a positive indicator of meaningful brain engagement. This suggests the fatigue reduction reflects improved energy management despite cognitive demands, rather than absence of cognitive exertion.

### Usability and technology acceptance

Perceived competence increased significantly across sessions, with participants requiring less technical assistance by post-assessment. This improvement is notable given the known barrier of technology anxiety in older populations and individuals less familiar with digital devices [47,48]. Interest/enjoyment, effort, value, and pressure subscales showed no significant changes, suggesting stable rather than increasing motivation [49]. This disconnect between objective performance gains and stable motivation scores may reflect the difficulty participants experienced in recognising subtle improvements.

All ten participants indicated they would recommend the intervention, linking willingness to perceived benefit and valuing any opportunity for improvement. Systems Usability Scale scores substantially exceeded industry averages [20,50], supporting appropriate design for this population. These findings align with evidence that adequate training and ongoing support can overcome technology-related barriers and empower participants to use technology-based rehabilitation tools independently [51].

### Study limitations

One important consideration is the time point of delivery in the stroke rehabilitation pathway. The cohort's heterogeneous time-since-stroke is an important consideration. Some participants felt they would have benefited more had the intervention been delivered earlier in recovery. Time since stroke moderated expectations, with those many years post-stroke expressing realistic expectations while recognising potential value for others. This observation aligns with evidence that visual impairments are often missed or deprioritised early in stroke recovery [2,11], and that earlier intervention delivery may facilitate greater neuroplasticity-dependent recovery [1,16]. Systematic investigation of optimal timing is warranted.

Future research should work to identify the optimal time point of delivery to ensure stroke survivors get the maximum benefit from engaging in visual rehabilitation as well as moving towards a more unified standard of care to ensure that visual impairments are identified early in recovery and appropriate support provided. Generally, knowledge in this area is scarce, specifically regarding effectiveness, demonstrating a need for further research [13]. Our study is a pilot study with a small sample, with limited statistical power and generalisability. While effect sizes for significant findings were moderate to large, findings should be interpreted cautiously. No control group was employed, preventing determination whether improvements resulted from the intervention or alternative factors (e.g., expectancy effects, repeated testing). Maintenance of benefits beyond post-intervention was not assessed. Therefore, future work should investigate the use of TVT in a larger cohort of stroke survivors. While technological visuo-cognitive training has been validated in other neurological populations (such as Parkinson's disease [20,29]), further stroke-specific validation is warranted to establish the evidence base for this intervention in post-stroke visual rehabilitation.

All intervention delivery and assessment occurred in a laboratory setting rather than in participants' home environments, limiting conclusions about the feasibility and efficacy of TVT for home-based rehabilitation. The laboratory context differs substantially from home environments in terms of environmental factors, technical support availability, and user autonomy. Importantly, independent home use requires participants to manage technical issues and maintain motivation without professional oversight, factors not directly assessed in this study. Future research should investigate TVT delivered in home settings, with particular attention to factors supporting adherence, user independence, and maintenance of benefits over extended periods.

High quality randomised-controlled trials would allow for full exploration of the acceptability, feasibility, and associated costs of technological interventions for visual rehabilitation and vision-related quality of life such as the SSS. Conducting this research could lead to insight into the benefits of compensatory training in a stroke population while providing a rigorous evidence base to inform clinicians and healthcare professionals.

## Conclusions

TVT appears to be a feasible and acceptable option for stroke rehabilitation with the potential to enhance visual search strategies and improve vision-related quality of life. The selective pattern of improvement, significant gains in visual search, patient-reported visual difficulties, and training-specific visuo-motor skills, suggests technology-based visual rehabilitation can target specific functional deficits relevant to daily activities [52]. High usability and increasing competence in our findings indicate technological barriers can be overcome with appropriate design.

Future work should aim to include larger samples stratified across the stroke recovery timeline and randomised controlled trials with control group comparison to determine whether TVT is superior to current rehabilitation methods. Research investigating optimal intervention duration, benefit maintenance, transfer to untrained domains, comparative effectiveness, and cost-effectiveness across varied clinical settings is needed to establish a rigorous evidence base for clinical implementation.

## Acknowledgments

We extend our thanks to all the participants who so willingly gave their time to be involved in this study. Their participation was crucial to the success of this project, and we are truly grateful for their generosity and commitment. The authors would also like to gratefully acknowledge Northumbria University for their generous Seedcorn funding, which was instrumental in making this project possible. This support provided the essential resources to conduct the research.

## Author contributions

**Conceptualization:** Samuel Stuart, Stephen Dunne.

**Data curation:** Abbey Fletcher, Beckie Morris, Stephen Dunne.

**Formal analysis:** Lewis Jefferson, Abbey Fletcher, Beckie Morris, Julia Das, Stephen Dunne.

**Funding acquisition:** Samuel Stuart, Stephen Dunne.

**Investigation:** Abbey Fletcher, Beckie Morris, Rosie Morris, Stephen Dunne.

**Methodology:** Julia Das, Stephen Dunne.

**Project administration:** Stephen Dunne.

**Supervision:** Stephen Dunne.

**Validation:** Samuel Stuart.

**Visualization:** Abbey Fletcher.

**Writing – original draft:** Abbey Fletcher, Julia Das, Rosie Morris, Samuel Stuart, Stephen Dunne.

**Writing – review & editing:** Lewis Jefferson, Abbey Fletcher, Beckie Morris, Julia Das, Rosie Morris, Samuel Stuart, Stephen Dunne.

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
