## [Decision Letter · Decision Letter 0]

30 Jun 2025

Response to Reviewers
Revised Manuscript with Track Changes
Manuscript
**Journal Requirements:**
**Additional Editor Comments (if provided):**

2.) Similarly, it’s not clear how the System Usability Scale could have been collected at baseline, given that participants would not have had experience with the intervention yet. So, they would be answering items like “I thought the system was easy to use” without having actually used the system? Please check.

3.) Clarity would be improved if the design subsection of the methods section more directly and clearly stated that this study is a single-arm pre-test, post-test design

4.) The fact that the follow-up assessment was not collected at a specific timepoint - authors should note this, as well as report descriptive statistics (including a range) of the time since baseline that follow-ups were collected

5.) I do not believe the Intrinsic Motivation Inventory assesses either usability or accessibility. What was this measure used to assess, then? In addition, what measures or approaches were used to assess feasibility and acceptability? Please clarify

6.) Authors should state what approach they used to qualitative analyses and include a reference. Was any other author involved in coding other than AF? Was agreement upon codes assessed? I believe most well-accepted approaches to qualitative analysis call for >1 coder.

7.) Whenever the means of two scores are compared, they should clearly identify which mean is pre-test and which is post-test, and include standard deviations for each. This applies to Table 1 & Table 2, as well.

8.) In the first paragraph of the Results section, the authors report that “there were no further significant differences found for other items.” The authors should specifically report each item they evaluated – focusing on only “significant” differences can make it seem as though the collected evidence is stronger, just by weight of emphasis.

9.) In the discussion, the authors conclude that the system was “usable,” which may be due to post-test SUS scores alone – but these raw scores do not appear to be reported. Please make sure these scores are reported.

**Reviewers' Comments:**

**Comments to the Author**

Reviewer #1: Summary:

The current manuscript details a pilot study assessing the useability and efficacy of a compensatory training tool for use in post-stroke visual rehabilitation. This is an important and under-studied field, addressing the needs of an underserved clinical population. While only a pilot study, and thus dealing with a small sample size and no control group, the authors provide good evidence that the proposed intervention is accessible for stroke survivors and provides benefit to daily living. Expanded discussion of the training device, patient details, and interpretation of improvements would improve the overall quality of the paper.

General comments:

The Discussion would benefit from expanded discussion of several factors, including the time post-stroke of the participants (reducing the potential confound of spontaneous recovery), as well as the validation of the Sensory Station in other disease models.

There is now substantial (if controversial) literature regarding visual rehabilitation training following stroke to improved quality of life and reduce the visual deficit. It is unclear where the SSS and TVT in general fit into this growing field.

Is there any proposed mechanism or theory behind the benefit of this training? While this is a limited pilot study, there is limited motivation beyond clinical benefit, which makes it unclear the real benefit of the intervention. Would stroke patients without visual impairments receive a similar benefit simply from performing cognitively demanding tasks and learning to interact with the technology? A unifying theory of why this training is expected to be beneficial would aid in interpreting the outcomes.

Specific comments:

SSS should be defined in the abstract.

The “Participants” heading under Results is uninformative and confusing, given the “Participants” heading listed again under Methods.

Formatting of Table 3 and Table 4 is causing some odd line overruns, making them a little hard to read.

It is unclear how “Impairment” is defined in Table 3 for each of the listed categories.

Reviewer #2: Overall Summary

This manuscript reports on a pilot study evaluating the feasibility and short-term effects of a technological visuo-cognitive training intervention using the Senaptec Sensory Station for stroke survivors with homonymous visual field loss. The study integrates quantitative (pre/post cognitive assessments) and qualitative (interviews) data to assess potential functional changes and system usability.

The topic is clinically relevant and aligns well with the scope of PLOS Digital Health, particularly in its exploration of novel rehabilitation tools for neurologically impaired populations. The methodology is broadly appropriate for a feasibility study, and the mixed-methods design is a strength. However, I believe several issues related to interpretation, statistical transparency, methodological rigor, and literature framing should be addressed to improve the manuscript’s clarity and credibility.

Major Comments:

1. Interpretation of key outcome measures (Fatigue & Falls Concern)

The paper reports on improvements in fatigue and fear of falling; however, as currently presented, the FES-I and FSS scores appear to increase post-intervention, which (according to standard scoring conventions) would indicate worsened outcomes. This may reflect a mislabelling or misordering of scores, but it is difficult to determine without “pre” and “post” labels. The directionality and scoring of these scales should be explicitly described and correctly interpreted in both the Results and Discussion sections.

2. Overstating efficacy in an uncontrolled, underpowered design

The manuscript uses language suggestive of clinical improvement or causal inference (e.g., “improved functioning,” “demonstrated efficacy”), which is not supported by the study’s design. The absence of a control group, the small and heterogeneous sample (n = 10; stroke onset 1–30 years), and the exploratory nature of the study limit the strength of any claims. More tentative phrasing (e.g., “observed changes” or “preliminary findings”) would be more appropriate throughout. The limitations section does briefly note that multiple comparisons were made, but it does not indicate whether any correction was applied or considered. Given the number of outcomes tested, this increases the risk of Type I error. If no correction was applied, the rationale should be clearly stated (e.g., due to the exploratory nature of the study). Additionally, the abstract highlights statistically significant findings using p < .05 values, without clarifying that these are uncorrected. This may overstate the robustness of the results. If uncorrected thresholds are reported, this should be clearly acknowledged.

3. Transparency and rigor in qualitative analysis

Thematic analysis is a central component of this mixed-methods study. However, it appears that coding was conducted by a single researcher, with no mention of inter-coder reliability, peer debriefing, or reflexivity. While adding a second coder retrospectively may not be feasible, the authors should (1) explicitly acknowledge this limitation, (2) add a brief reflexivity statement regarding the coder’s background and potential biases, and (3) describe any procedures used to support the trustworthiness of the analysis (e.g., consultation with co-authors or iterative team discussions). These steps would strengthen confidence in the qualitative findings.

4. Limited engagement with relevant literature on digital interventions for hemianopia: While the authors reference one prior study involving a digital tool for visual rehabilitation, the discussion of related work remains narrow. Key studies such as those related to the Eye-Search project (e.g., Leff et al., 2020) are not included, nor are feasibility studies involving gamified interventions for children with hemianopia (e.g., Ivanov et al., 2018; Waddington et al., 2018). Although these studies primarily focus on oculomotor training (unlike the cognitive training approach used in this study) they establish important precedents for the feasibility, engagement, and translational potential of digital tools in visual rehabilitation. Broadening the literature review and discussion to reflect this wider body of work would help position the current intervention more accurately within the landscape of digital rehabilitation research for hemianopia.

Minor comments:

Senaptec System Validity: The manuscript does not address the psychometric properties of the Senaptec Sensory Station tasks, which form the basis of the intervention and outcome measurement. While this is acceptable in a pilot context, the authors should clarify whether these tasks are validated neuropsychological measures or proprietary tools without established reliability or construct validity. A brief acknowledgment one way or the other would enhance transparency.

Cognitive Inclusion Criteria: The paper states participants had "good cognitive ability" but does not clarify how this was determined. Please clarify whether a formal screening tool or clinical judgment was used.

Terminology Clarification: The use of the term “pseudo-random” to describe task ordering is unclear. Please indicate whether this refers to a fixed sequence, partial counterbalancing, or some other approach.

Abstract and Introduction: Some sentences are verbose or repetitive. For example, “post-stroke visual field loss is a common impairment after stroke” could be condensed. Consider revising for conciseness and clarity.

Motivational Measures: The Intrinsic Motivation Inventory is an appropriate tool, but the version used and any adaptations made for this population should be described.

Table Formatting: Statistical tables should be checked for completeness and alignment with journal expectations. In particular, they should include degrees of freedom, and ideally effect sizes and confidence intervals, to support interpretability.

This is a promising study with the potential to contribute to digital rehabilitation literature. Addressing the points above, especially around the interpretation of outcome measures, the tone of efficacy claims, and the robustness of qualitative analysis, will substantially strengthen the manuscript.

**Do you want your identity to be public for this peer review?** For information about this choice, including consent withdrawal, please see our Privacy Policy

Reviewer #1: No

Reviewer #2: No

**Figure resubmission:****Reproducibility:** To enhance the reproducibility of your results, we recommend that authors of applicable studies deposit laboratory protocols in protocols.io, where a protocol can be assigned its own identifier (DOI) such that it can be cited independently in the future. Additionally, PLOS ONE offers an option to publish peer-reviewed clinical study protocols. Read more information on sharing protocols at https://plos.org/protocols?utm_medium=editorial-email&utm_source=authorletters&utm_campaign=protocols

---

## [Editor Report · Decision Letter 1]

6 Feb 2026

Trialling the efficacy of a technological visuo-cognitive training program as a compensatory tool for visual rehabilitation after stroke: A pilot study

PDIG-D-25-00088R1

Dear Dr Dunne,

We are pleased to inform you that your manuscript 'Trialling the efficacy of a technological visuo-cognitive training program as a compensatory tool for visual rehabilitation after stroke: A pilot study' has been provisionally accepted for publication in PLOS Digital Health.

Best regards,

Tyler Wray

Academic Editor

PLOS Digital Health